# Antimicrobial Properties of *Lepidium sativum* L. Facilitated Silver Nanoparticles

**DOI:** 10.3390/pharmaceutics13091352

**Published:** 2021-08-27

**Authors:** Samir Haj Bloukh, Zehra Edis, Hamid Abu Sara, Mustafa Ameen Alhamaidah

**Affiliations:** 1Department of Clinical Sciences, College of Pharmacy and Health Science, Ajman University, Ajman P.O. Box 346, United Arab Emirates; s.bloukh@ajman.ac.ae (S.H.B.); h.abusara@ajman.ac.ae (H.A.S.); mou.94.95@hotmail.com (M.A.A.); 2Center of Medical and Bio-Allied Health Sciences Research, Ajman University, Ajman P.O. Box 346, United Arab Emirates; 3Department of Pharmaceutical Sciences, College of Pharmacy and Health Science, Ajman University, Ajman P.O. Box 346, United Arab Emirates

**Keywords:** antibiotic resistance, antimicrobial resistance, *Lepidium sativum* L., silver nanoparticles, biomaterials, antimicrobial activity, synergism, green synthesis, surgical site infection, wound dressing

## Abstract

Antibiotic resistance toward commonly used medicinal drugs is a dangerously growing threat to our existence. Plants are naturally equipped with a spectrum of biomolecules and metabolites with important biological activities. These natural compounds constitute a treasure in the fight against multidrug-resistant microorganisms. The development of plant-based antimicrobials through green synthesis may deliver alternatives to common drugs. *Lepidium sativum* L. (LS) is widely available throughout the world as a fast-growing herb known as garden cress. LS seed oil is interesting due to its antimicrobial, antioxidant, and anti-inflammatory activities. Nanotechnology offers a plethora of applications in the health sector. Silver nanoparticles (AgNP) are used due to their antimicrobial properties. We combined LS and AgNP to prevent microbial resistance through plant-based synergistic mechanisms within the nanomaterial. AgNP were prepared by a facile one-pot synthesis through plant-biomolecules-induced reduction of silver nitrate via a green method. The phytochemicals in the aqueous LS extract act as reducing, capping, and stabilizing agents of AgNP. The composition of the LS-AgNP biohybrids was confirmed by analytical methods. Antimicrobial testing against 10 reference strains of pathogens exhibited excellent to intermediate antimicrobial activity. The bio-nanohybrid LS-AgNP has potential uses as a broad-spectrum microbicide, disinfectant, and wound care product.

## 1. Introduction

Antibacterial resistance is according to the WHO causing increased morbidity and mortality throughout low- and medium-income countries due to a widespread, uncontrolled use of antibacterial agents [1]. The WHO report of 2020 “Prioritization of Pathogens to Guide Research and Development of New Antibiotics” mentions several pathogens as critical priority including the carbapenem-resistant *Pseudomonas aeruginosa* [1]. *Staphylococcus aureus* is listed as vancomycin and methicillin-resistant Gram-negative bacteria under the high-priority pathogens [1]. *Streptococcus pneumoniae* is not susceptible to penicillin and is considered a medium-priority pathogen [1]. Nosocomial infections acquired in hospital settings due to ESKAPE (*Enterococcus feacium*, *S. aureus*, *Klebsiella pneumoniae*, *Acinetobacter baumannii*, *P. aeruginosa*, *Enterobacter* species) pathogens lead to rising cases of morbidity and mortality everywhere in the world [2]. Increased treatment costs, duration, and delayed or no recovery are the outcomes of resistance [2,3,4]. Resistance against antibiotics during COVID-19 resulted in increased fatalities and serious setbacks [2,3,4]. Plant-based biomolecules and nanomaterials offer promising alternatives [2,5,6,7,8,9,10,11]. The use of carboxylic acids, polyphenols, and further phytochemicals against the coronavirus is discussed in recent papers [8,9,10,11]. Plant phytochemicals act also as reducing, stabilizing, and capping agents for bio-synthesized nanoparticles [12]. Nanoparticles can be prepared by green methods and are utilized as antimicrobials or drug carriers [2,12,13,14]. The mechanisms of antimicrobial activity of silver nanoparticles (AgNP) and gold nanoparticles were investigated in recent papers [15,16,17]. AgNP and silver ions may coexist and inhibit especially Gram-negative bacteria [18,19,20,21]. Once released by AgNP, they enter across the bacterial outer membrane through porin channels and ion transporter proteins [18]. Silver ions can cause oxidative stress and DNA strand damage through interaction with proteins [18]. Other mechanisms involve inhibiting proteins synthesis and DNA replication by interacting with ribosomal proteins and nucleic acids, respectively [18]. The mechanisms of antimicrobial inhibition by AgNP are not yet fully explained [18]. AgNP with sizes smaller than 17 nm can cause cell death by puncturing the cell membranes according to El-Kahky et al. [13]. Many factors determine the effectivity of AgNP against microorganisms [18]. The size of AgNP is determined by the stability and adsorption pattern of coating/capping agents on the Ag surface [18]. Strong reducing agents result in faster reaction rates, smaller, more stable NP with a narrow size distribution, and better microbial inhibition [17]. An agglomeration of AgNP reduces antimicrobial properties and is induced by the storing, aging, or degradation of the capping agent [18]. The size of AgNP depends on the pH level, concentration of silver ions, effectivity, and synergistic mechanisms of the available phytochemicals during the green synthesis [18,19,20,21,22,23,24,25,26,27,28,29,30,31,32,33,34,35,36,37]. The plant constituents can enhance the antimicrobial properties of the AgNP [17].

Medicinal/herbal plants are sources of phenolic acids, polyphenols, flavonoids, terpenoids, and further phytochemicals [3,6,8]. Phytochemicals can reduce biofilm formation, inhibit quorum sensing, prevent bacterial attachment on mucosal surfaces, influence cell surface hydrophobicity, and reduce glycolytic enzymes [23,24]. The antimicrobial activity of plant constituents is ruled by the morphology and structure of the target pathogen. This finding is in agreement with our previous investigations on complexes of “smart” triiodides, which were produced with the addition of molecular iodine (I_2_) [37,38,39,40,41]. We utilized plant extracts of *Capsicum frutescens* (Paprika), *Zingiber officinale* (Ginger), *Aloe vera barbadensis* M. (AV), *Cinnamomum zeylanicum* (Cinn), and *Salvia officinalis* L. (Sage) in different formulations with AgNP and/or iodine [25,42,43,44]. The plant extracts contain *trans*-cinnamic acid (TCA), ferulic acid, caffeic acid, rosmarinic acid, carnosic acid, sagerinic acid, salvianolic acid, sage coumarin, and many other biocomponents [42,43,44,45,46,47,48,49,50,51]. These phytochemicals are also constituents in curly garden cress, which is also called *Lepidium sativum* L. (LS) [52].

Cress (LS) is a member of the *Brassicaceae* species (Cruciferae family) and is known as garden cress. LS is a fast-growing herbal plant, which grows in many regions of the world [52,53,54,55,56,57,58,59,60,61]. Many papers reported antimicrobial, antioxidant, and anti-inflammatory activities of LS seed oil [52,53,54,55,56,57,58,59,60,61,62]. The whole plant of LS contains cinnamic-, benzoic-, salicylic-, gallic-, ferulic-, caffeic-, *p*-coumaric-, chlorogenic-, vanilic acid, polysaccharides, and further biocompounds [54,57]. We encapsulated AgNP with LS extract to ameliorate the microbicidal activities. The nano-biohybrid LS-AgNP may prevent microbial resistance and biofilm formation by plant-based synergistic mechanisms [22,23,24]. Such properties are relevant for the treatment of wounds [63]. Effective antimicrobial agents help in faster wound closure, and they also reduce the time and costs of wound-treatment regimens [42,64]. The inability to control wound infections due to resistance and biofilm formation can result in chronic infections or even death [63]. Our investigations on polyvinylpyrrolidone (PVP) and povidone iodine (PI) stabilized TCA-AgNP, TCA-AgNP-PI, Cinn-AgNP, Cinn-AgNP-PI, AV-PVP-I_2_, AV-PVP-I_2_-NaI, and AV-PVP-Sage-I_2_ showed promising results in disk diffusion studies [42,43,44]. The Sage biohybrid exhibited higher inhibitory action against *Candida albicans* and the Gram-positive bacteria compared to all the other formulations [42,43,44]. These studies confirm the antimicrobial action of phytochemicals in AV, Sage, and Cinn extracts [42,43,44]. The concerned plant constituents are TCA and cinnamaldehyde for the Cinn and TCA nanocolloids [43]. The AV-PVP-Sage-I_2_ biohybrid consists mainly of sagerinic acid, salvianolic acid, sage coumarin, trans-rosmarinic acid, ferulic acid, vanilic acid, and caffeic acid [43]. The AV extract contains acemannan and cinnamic acid [43].

Relying on our previous investigations on plant constituents in “bio-antimicrobial” agents, we explored the effect of LS in combination with AgNP. We aimed to enhance the inhibitory activities of AgNP by utilizing LS as a reducing, stabilizing, and capping agent in different concentrations (5 and 10%) and pH levels (pH 7 and 8.5) in a time-dependent manner (fresh until 1 week old). We used these small concentrations in order to reduce the amount of AgNP within the biohybrid due to concerns related to bacterial resistance, cytotoxicity, environmental damage, and cost-effectiveness [65,66,67,68,69,70]. AgNP can initiate bacterial resistance to antibiotics through enhancing bacterial stress tolerance, which can be acquired by previous exposure to AgNP [18,65,68]. Another publication indicates rising AgNP and silver ion resistance of ESKAPE pathogens due to co-selection [67]. The exposure may be due to the manifold uses of AgNP in healthcare, consumer products, cosmetics, food packaging, textiles, and further uses in industry, as well as in agriculture [31,33,65,68]. Other concerns are due to environmental damage and cytotoxicity inflicted on living systems by AgNP [13,69]. These disadvantages of AgNP and Ag-ions encouraged us to utilize LS extract-mediated biosynthesis of AgNP with a low concentration of silver. Safety, sustainability, cost-effectiveness, and facile preparation by green synthesis are our main goals during our investigation of antimicrobial formulations of LS-AgNP.

AgNP were prepared in a one-pot synthesis by LS plant-biomolecules-induced reduction of silver nitrate via a green method. The biomolecules and metabolites in the aqueous LS extract act as reducing, capping, and stabilizing agents of AgNP. Scanning Electron Microscopy (SEM) and Energy-Dispersive X-ray Spectroscopy (EDX) were utilized to analyze the morphology, composition, and distribution of LS-AgNP. Fourier transform infrared spectroscopy (FT-IR), Ultraviolet-visible spectroscopy (UV-Vis), and X-ray Diffraction (XRD) confirmed the composition of the LS-AgNP biohybrids. Dynamic light scattering (DLS) was used to determine the size distribution and stability of the nanoparticles. Antimicrobial testing by the disc dilution method against a total of 10 reference strains of microorganisms verified excellent to intermediate antimicrobial activity. The Gram-negative pathogens *Escherichia coli* WDCM 00013 and *P. aeruginosa* WDCM 00026 were highly inhibited, which was followed by intermediate results for the Gram-positive bacteria *Bacillus subtilis* WDCM 00003, *S. pneumoniae* ATCC 49619, *S. aureus* ATCC 25923, *Streptococcus pyogenes* ATCC 19615, *E. faecalis* ATCC 29212, and the fungus *C. albicans* WDCM 00054. Our biohybrid LS-AgNP showed increased antimicrobial activity with potential uses as a disinfectant and wound care product.

## 2. Materials and Methods

### 2.1. Materials

Sabouraud Dextrose broth, Mueller–Hinton Broth (MHB), and silver nitrate were purchased from Sigma Aldrich (St. Louis, MO, USA). The microbial strains *E. coli* WDCM 00013 Vitroids, *P. aeruginosa* WDCM 00026 Vitroids, *K. pneumoniae* WDCM 00097 Vitroids, *C. albicans* WDCM 00054 Vitroids, and *Bacillus subtilis* WDCM 0003 Vitroids were purchased from Sigma-Aldrich Chemical Co. (St. Louis, MO, USA). Liofilchem (Roseto degli Abruzzi, Teramo, Italy) delivered *S. pneumoniae* ATCC 49619, *S. aureus* ATCC 25923, *E. faecalis* ATCC 29212, *S. pyogenes* ATCC 19615, and *P. mirabilis* ATCC 29906. Disposable sterilized Petri dishes with Mueller–Hinton II agar, McFarland standard sets, gentamicin (9125, 30 µg/disc), and nystatin (9078, 100 IU/disc) were obtained from Liofilchem Diagnostici (Roseto degli Abruzzi, Teramo, Italy). Sterile filter paper discs with 6 mm diameter were obtained from Himedia (Jaitala Nagpur, Maharashtra, India). Curly garden cress (*Lepidium sativum* L., (LS)) seeds were purchased from the local market. The seeds were planted and harvested during November 2020 to April 2021. Ultrapure water was used in the extraction and green synthesis instead of distilled water. All reagents were used as delivered and were of analytical grade.

### 2.2. Preparation of LS Extract

We obtained the seeds of LS from a local shop in Sharjah, UAE and planted them in November 2020. The plants grew up to 12 cm and were harvested from December 2020 to April 2021 in the morning hours. They were immediately transported to the research lab of the College of Pharmacy and Health Sciences, Ajman University, Ajman, UAE. The LS were rinsed several times with water to remove soil and impurities. Then, they were washed thoroughly three times with distilled water and finally with ultrapure water. The plants were left for drying at ambient temperature for 2 h until the remaining water evaporated. The extraction method was adapted partly from previous investigations [28,36]. 10 g of LS whole plants were added into a beaker with 100 mL of ultrapure water and heated to 60 °C. The covered beaker was kept for 30 min at 60 °C under continuous stirring. The green solution was allowed to cool down to room temperature. Then, 10 mL of LS extract was diluted with ultrapure water to 100 mL. For the investigation of the pH dependency, a few drops of 10% diluted NaOH solution were added until the pH was adjusted to 8.5 under continuous stirring. The other stock solution was adjusted to pH = 7. The light green color changed to greenish-yellow due to the dilution and change of pH. These stock solutions were immediately used for the preparation of LS-AgNP.

### 2.3. Preparation of LS-AgNP

Silver nitrate solution was prepared by adding 0.183 g (1.077 mmol) AgNO_3_ into 100 mL of ultrapure water at 0 °C and stirring for 10 min. Then, 5 and 10 mL of the silver nitrate solution were added separately to the prepared LS stock solutions under continuous stirring for 30 min at 60 °C. The final products changed from greenish-yellow to yellowish-brown color. The colloids with the added silver nitrate concentrations of 5% (0.5386 μg/mL) and 10% (1.0773 μg/mL) were kept at 3 °C until further use.

### 2.4. Characterization of LS-AgNP

The nanocomposite LS-AgNP was analyzed by SEM/EDS, UV-vis, and FTIR, and X-ray diffraction (XRD). The results confirmed the composition and purity of LS-AgNP.

#### 2.4.1. Scanning Electron Microscopy (SEM) and Energy-Dispersive X-ray Spectroscopy (EDX)

The scanning electron microscopy (SEM, VEGA3, Tescan, Brno, Czech Republic) analysis of LS-AgNP was used to study the morphology of the nanocomposite LS-AgNP. The elemental composition of our sample was investigated by energy-dispersive X-ray spectroscopy (EDS, VEGA3, Tescan, Brno, Czech Republic). The analysis was performed on VEGA3 from Tescan (Brno, Czech Republic) at 20 kV. The same instrument was used for the EDS analysis. The measurements were done by dispersing one drop of LS-AgNP into distilled water and placing this suspension onto a carbon-coated copper grid. After drying the sample under ambient conditions, it was coated by a Quorum Technology Mini Sputter Coater (Quorum Technologies, Laughton, East Sussex, UK) with a gold film.

#### 2.4.2. Size and Zeta Potential Analysis

The calculation of the average size, size distribution, and the polydispersity index (PDI) was performed by dynamic light scattering (DLS) analysis with the model SZ-100 purchased from Horiba (Palaiseau, France). Dispersion and stability of the nanocolloidal biohybrid was undertaken at room temperature by zeta (ζ) potential measurement.

#### 2.4.3. UV-Vis Spectrophotometry (UV-Vis)

The UV-Vis spectrophotometer model 2600i from Shimadzu (Kyoto, Japan) was utilized at a wavelength range from 195 to 800 nm in the analysis of LS-AgNP.

#### 2.4.4. Fourier Transform Infrared Spectroscopy (FTIR)

An ATR IR spectrometer with a Diamond window from Shimadzu (Kyoto, Japan) was used for the FTIR analysis of LS-AgNP. The nanocomposite was freeze-dried and measured within the wavenumber range of 400 to 4000 cm^−1^.

#### 2.4.5. X-ray Diffraction (XRD)

A Bruker XRD (BRUKER, D8 Advance, Karlsruhe, Germany) was used to study the nanomaterial LS-AgNP. The Cu radiation had a wavelength of 1.54060 A. The analysis was done by coupled Two Theta/Theta with a time-step of 0.5 s and step size of 0.03.

### 2.5. Bacterial Strains and Culturing

The antimicrobial properties of LS-AgNP were studied against the reference strains of the pathogens *S. pneumoniae* ATCC 49619, *S. aureus* ATCC 25923, *E. faecalis* ATCC 29212, *S. pyogenes* ATCC 19615, *Bacillus subtilis* WDCM 0003 Vitroids, *P. mirabilis* ATCC 29906, *E. coli* WDCM 00013 Vitroids, *P. aeruginosa* WDCM 00026 Vitroids, *K. pneumoniae* WDCM 00097 Vitroids, and *C. albicans* WDCM 00054 Vitroids. These microorganisms were stored at −20 °C. MHB was inoculated by adding the fresh microbes and was kept at 4 °C until needed in the study.

### 2.6. Determination of Antimicrobial Properties of LS-AgNP

The antimicrobial properties of our title bio-nanocomposite were studied against nine reference bacterial strains and one fungal strain. The bacterial strains consisted of Gram-positive *S. pneumoniae* ATCC 49619, *S. aureus* ATCC 25923, *S. pyogenes* ATCC 19615, *E. faecalis* ATCC 29212, and *B. subtilis* WDCM 00003. Gram-negative bacteria were *P. mirabilis* ATCC 29906, *P. aeruginosa* WDCM 00026, *E. coli* WDCM 00013, and *K. pneumoniae* WDCM 00097. The antibiotic gentamycin was used as positive control for all the bacterial strains. The fungal reference strain *C. albicans* WDCM 00054 was compared to the antibiotic nystatin. Water was used as solvent and as negative control. The negative controls showed no inhibition zones and were not included in the discussion. All the antimicrobial tests were performed thrice, and their average was used in this investigation.

#### 2.6.1. Zone of Inhibition Plate Studies

The microbial susceptibility against our nanocompound LS-AgNP was tested by zone of inhibition plate studies [70]. The selected bacterial strains were added into 10 mL of MHB and kept at 37 °C for an incubation period of 2 to 4 h. We used Sabouraud Dextrose broth to incubate *C. albicans* WDCM 00054 at 30 °C. Sterile Petri dishes with MHA were evenly seeded by sterile cotton swabs with a selected microbial culture of 100 μL conform to 0.5 McFarland standard. After a drying period of 10 min, the plates were ready for antimicrobial studies.

#### 2.6.2. Disc Diffusion Method

We followed the descriptions of the Clinical and Laboratory Standards Institute (CLSI) to test the susceptibility of our 10 reference strains. We used the common antibiotic discs of nystatin and gentamycin as comparison [71]. Two mL of our nanocolloids with the 5% (0.54 µg/mL, 0.27 µg/mL, 0.14 µg/mL) and 10% (1.08 µg/mL, 0.54 µg/mL, 0.27 µg/mL) concentrations were used to impregnate sterile filter paper discs for 1 day. The soaked disks were dried for 24 h under ambient conditions. We incubated the fungal reference strain *C. albicans* WDCM 00054 on agar plates at 30 °C for 24 h. We used a ruler to measure the clear zone of inhibition (ZOI) diameter around the disc to the nearest millimeter. If there is no inhibition zone, the reference strain is not susceptible. In this case, the ZOI is equal to zero and the strain is resistant.

### 2.7. Statistical Analysis

The statistical analysis was done by SPSS software (version 17.0, SPSS Inc., Chicago, IL, USA). Mean values are presented as data, and one-way ANOVA was employed to calculate the statistical significance between groups. Values with *p* < 0.05 were noted as statistically significant.

## 3. Results and Discussion

The incidence of microbial resistance against common drugs and antimicrobial agents is a matter of concern. The future of mankind is at stake due to the notorious ESKAPE pathogens [1,2]. Medicinal plants may be an important alternative in the battle against resistant microorganisms [2,3,5,6,7,8,9,10,11]. Medicinal plants use synergistic mechanisms based on their antimicrobial constituents [3,5,6,7,8,9,10,11,23,24]. Plant constituents can inhibit biofilm formation, protein synthesis, and quorum sensing [3,5,6,7,8,9,10,11,23,24]. Silver ions and AgNP have similar antimicrobial properties [2,12,13,14,15,16,17,18,19,20,21]. Recent investigations on the green synthesis of AgNP gained popularity. Green methods are based on simple, cost-effective methods and sustainable, eco-friendly materials [15]. The use of biomass, waste materials, and plant constituents as reducing/stabilizing/capping agents is part of bio-silver nanoparticle synthesis [15]. Such methods encourage the development of nano-bio-antimicrobials, which can be used against resistant pathogens. Based on our experience on bio-synthesized antimicrobials, we investigated the use of curly garden cress (LS) extract. LS biocomponents acted as reducing and stabilizing agents in the green synthesis of AgNP. The facile one-pot synthesis of LS-AgNP produced cost-effective, sustainable antimicrobial agents in an eco-friendly way. The biohybrid LS-AgNP revealed pH, concentration, and time dependency. The alkaline pH of 8.5 during the synthesis produced stable LS-AgNP with a smaller size distribution and better antimicrobial activities. The same results are achieved when 10% silver ion concentration is used instead of 5%. The stability and antimicrobial action of LS-AgNP are inversely related to aging and decreased in a time-dependent pattern during storage after 7 days. The highest inhibitory action against the tested microorganisms was achieved by the 10% LS-AgNP prepared at pH of 8.5 with a concentration of 1.0773 μg/mL within the first 6 days. The UV-vis analysis supports this through a sharper surface plasmon peak for AgNP at λ-max = 447 nm. If not mentioned further, the analytical results are related to the 10% LS-AgNP colloid synthesized at a pH of 8.5 not older than 6 days.

### 3.1. Morphological Examination, Elemental Composition

#### 3.1.1. Electron Microscope (SEM) and Energy-Dispersive X-ray Spectroscopic (EDS) Analysis

The composition, purity, and morphology of LS-AgNP were investigated by SEM and EDS (Figure 1).

The SEM analysis in Figure 1a depicts a microcrystalline, heterogeneous morphology. The purity of this sample is confirmed by the EDS analysis (Figure 1c). The sample LS-AgNP consists of 83.6% Ag, 11.6% C, 3.4% O, and 1.4% Cl (Figure 1c). The results confirm the purity of LS-AgNP. The layered EDS shows a homogenous presence of Ag in all the sample (Figure 1b,d). Carbon and oxygen are also uniformly distributed in the sample (Appendix A). Cl ions are available in plant materials and form AgCl during the biosynthesis of AgNP [18,21,42]. Silver ions are released due to the oxidation of AgNP, moisture, or simply due to the equilibrium process [29,42]. The formation of a AgCl secondary phase was also observed during the preparation of Cinn-AgNP-PI in our previous investigations [42].

#### 3.1.2. Dynamic Light Scattering (DLS) and Zeta Potential Analysis

The DLS analysis confirms the unimodal size distribution of LS-AgNP, which is in agreement with the SEM results (Figure 1). Table 1 shows the results of the DLS measurements.

The zeta (ζ) potential of 10% LS-AgNP shows a small negative charge of −0.2 mV (Table 1).

The zeta potential is an indicator of the stability of a colloidal suspension. The small negative charge of −0.2 mV reveals rapid coagulation of the 10% LS-AgNP biohybrid. The PDI value highlights the size distribution of the molecules within the given sample. The 5% LS-AgNP has a higher PDI compared to the 10% sample (Table 1). This result indicates a broader size distribution in the 5% LS-AgNP sample due to agglomeration. The 10% LS-AgNP biohybrid shows a smaller PDI of 0.388 due to its smaller size distribution and less agglomeration (Table 1). The 10% LS-AgNP sample contains more stable, smaller, and polydisperse AgNPs. The Z-average for the 5% and the 10% LS-AgNP samples are 65.5 and 57.9 nm, respectively (Table 1). The Z-average for the 10% LS-AgNP is smaller compared to the 5% sample and confirms the same findings of the PDI values (Table 1). The particle mean sizes of the 5% LS-AgNP is smaller than the 10% LS-AgNP sample, but the PDI of the latter is more favorable regarding stability, size distribution, and agglomeration.

The DLS analysis of the 10% LS-AgNP biocolloid indicates a homogenous size distribution around 36.1 nm (Appendix A). In the spectrum, there are no other peak intensities available. In other reports, the prevalence of bulky aggregates with a wide size distribution is observed [30,32]. The low polydispersion index of 0.388 confirms the presence of a uniform dispersion of LS-AgNP. The phytochemicals in the LS extract acted effectively as reducing and stabilizing agents, resulting in small, homogenous, and narrow size distribution [32]. Zeta (ζ) potential analysis renders further information about AgNP stability, size distribution, and agglomeration profiles in solution. The ζ-potential value of LS-AgNP is −0.2 mV and indicates a negatively charged AgNP surface (Table 1). The results verify the presence of a stable, small sized, homogenous biocolloid LS-AgNP. There are no aggregates, because the biomolecules within LS rapidly reduced the silver ions to Ag nanoparticles and prevented a secondary nucleation on the silver surface by forming a monolayer [15,17]. Silver ions were reduced by the hydroxyl and carbonyl moieties within the biomolecules of LS consisting of phenolic acids, polyphenols, flavonoids, terpenoids, and proteins [15,54,56,57]. The monolayer of plant biomolecules counteracts agglomeration and improves the antimicrobial activities of AgNP [15,17].

### 3.2. Spectroscopical Characterization

The composition of the LS-AgNP can be elucidated by spectroscopical characterization. We used UV-vis, FT-IR, and X-ray diffraction techniques.

#### 3.2.1. UV-Vis Spectroscopy

UV-vis spectroscopy can be utilized to assess the formation and size of AgNP in the biocolloidal LS-AgNP. The presence of different biocompounds originating from the LS extract can be elucidated. The synthesis of the biocomposite LS-AgNP was analyzed by UV-vis spectroscopy at two different concentrations and times (Figure 2).

AgNP formation is confirmed in UV-vis analysis with a plasmonic peak in the range of around 400–500 nm (Figure 2). The peak is available at λ-max = 447 nm in the 10% LS-AgNP sample (Figure 2, red curve). After 7 days, the 10% LS-AgNP sample reveals a lower intensity band around 435 nm for AgNP (Figure 2, light blue curve). The lower intensity absorption after one week infers an increased encapsulation of AgNP by the biomolecules. Chromophores cannot absorb due to the lower availability of π-electrons. The molecular structure of LS-AgNP is coiled, resulting in a smaller size. Both of the 5% LS-AgNP formulations show a lower intensity small, broad band at around 412 nm (Figure 2). This broad peak comes in form of a shoulder with lower absorption intensity and is blue-shifted compared to the 10% sample. The blue shift from 447 to 412 nm and the hypochromic effect indicate a smaller NP size (Appendix A). Lower intensity absorptions indicate the adsorption of an increased number of biomolecules on the AgNP. This reduces the size of the AgNP by encapsulating the primary Ag layer and preventing a secondary layer of attached Ag nuclei [15,17]. The sharpness of the peak in the 10% LS-AgNP indicates a faster reduction rate of silver ions compared to the 5% formulation. The doubling of silver ion concentration from 5% to 10% resulted in a red shift and slightly bigger sizes of the NP in the latter. The DLS analysis confirmed the findings. The bathochromic effect means an increase in conjugated systems, chromophores, and solvent effect due to the phytochemicals in 10% LS-AgNP. The red shift indicates a change in the phytochemicals from C–O bonds to C=O and C–C to C=C. This leads to inner-particle interactions such as repulsion, steric hindrance, and crowding between adsorbed biomolecules through their conjugated, bulky phenolic groups on the surroundings of the Ag surface [42]. As a result, AgNP in 10% LS-AgNP are more stable because agglomeration is prevented by inner-particle interactions.

In the green synthesis of AgNP by *Cinnamomum zeylanicum* (Cinn) bark extract and trans-cinnamic acid (TCA), the UV-vis analysis showed surface plasmon peaks at 390–415 nm (broad) and 400 nm (sharp), respectively [42]. Soni et al. mentioned a broad peak at 480 nm for AgNP biosynthesized by *Cinnamomum zeylanicum* [34]. These results infer the availability of cinnamic acid and its derivatives caffeic acid, rosmarinic acid, coumaric acid, chicoric acid, ferulic acid, and further polyphenols [8,42]. AgNP bio-synthesis is a complex phenomenon and has an impact on the antimicrobial properties of the biocolloid. The best results in this work were achieved by the AgNP synthesis at pH 8.5 with an initial silver ion concentration of 10% at pH 8.5 (Figure 2, red curve).

The entity of the biomolecules responsible for the reduction of silver ions can be identified by UV-vis-spectroscopic analysis through comparative observation of absorption peaks. In general, phenolic compounds appear in the region around 320–380 nm and flavonoids appear in the region around 280–315 nm. These compound classes are represented in the green curve depicting the LS-extract (Figure 2). In Figure 2, the LS extract shows broad absorption intensities between 230 and 290 nm and 295–400 nm related to flavonoids and phenolic compounds, respectively.

This is in agreement with our investigations on TCA-AgNP, Cinn-AgNP, as well as TCA-AgNP-, Cinn-AgNP-, AV-PVP-, and AV-PVP-Sage-iodine formulations [42,43]. Pure TCA absorbs at 265 nm, which suggests the presence of trans-cinnamic acid in the LS-extract [42]. The broadness of the absorption infers the availability of other hydroxycinnamic acid derivatives as well. Pure Sage biomolecules show absorptions around 200–230 nm (high intensity), 260–300 nm (broad shoulder), as well as low-intensity absorption bands at 310–480 nm and 400–450 nm [43,45,50]. The spectrum of AV-PVP-Sage reveals strong absorption peaks at 283 and 338 nm, which are Sage and AV phenolic compounds [43,44,45]. The LS extract shows comparable broad bands around 200–220 nm, 240 to 290 nm (λ-max = 257 nm), and 300 to 350 nm (λ-max = 322 nm) (Figure 3). The two latter broad absorption bands belong to trans-rosmarinic acid and caffeic acid, which appear at 280/330 nm and 328 nm, respectively [43,50]. We reported in our AV-PVP-iodine formulations a peak at 331 nm originating from caffeic acid and rosmarinic acid previously [43]. The absorption at around 270 nm in pure LS may be due to rosmarinic, ferulic, caffeic, carnosic, and cinnamic acid [43,47]. The broad band around 320 nm suggests the availability of quercetin, chlorogenic acid, chicoric acid, *p*-coumaric acid, catechin, kaempferol, cirsimaritin, apigenin, luteolin, hesperidine, and thymol [8,26,50].

The UV-vis absorption pattern of pure LS changes, when silver ions are added into the plant extract. A red shift is seen around 200–220 nm toward 205–235 nm in the 10% LS-AgNP formulation (Figure 2, red curve). All the spectra show the same red shift except the 7-day-old 5% LS-AgNP (Figure 2, black curve). The biomolecules undergo a change to higher conjugation with more absorption by chromophores from C–O to C=O. This can be an indicator for the reduction of silver ions to AgNP. The reduction happens by the deprotonated hydroxyl-groups within the polyphenols in the slightly alkaline solution at pH 8.5 [26]:Ag^+^ + R–C–O^−^ ⇆ Ag + R–C=O (1)
with a high-intensity absorption at 219 and a shoulder at 228 nm. The related absorptions were available at 200 and 220 nm in the LS extract. The same reduction is indicated by the blue shift from 257 nm in the pure LS extract toward 248 nm in all the other spectra (Figure 2). The LS-bio-constituents absorbing at this range are rosmarinic, ferulic, caffeic, carnosic, and cinnamic acid. All phenolic acids except cinnamic acid contain hydroxyl groups and reduce the silver ions, as indicated in Equation (1). The reduction decreases the availability of π-electrons and reduces the conjugation in the ring systems. This is reflected in the lower absorption intensity of the older LS-AgNP biohybrids. The lower absorption intensity indicates higher encapsulation, more hydrogen bonding, and less availability of the chromophores within the phenolic acids. After 6 days and above, phenolic acids are fixed within an extended molecular network due to inner-particle interactions. The fresher samples of 5% and 10% LS-AgNP show up to 6 days higher absorption intensities from 230 to 800 nm compared to the pure LS sample. The stability and antimicrobial properties are directly related to the availability of conjugated systems, free π-electrons, and less hydrogen bonding. The two samples contain intact phenolic acids (rosmarinic, ferulic, caffeic, carnosic, and cinnamic acid), which stabilize the AgNP by capping. The capping counteracts agglomeration of the AgNP [42]. In this process,
AgNP + R–C–O^−^ ⇆ R–C–O^−^⋯⋯AgNP (2)
where negatively charged oxygen atoms originating from carbonyl groups and deprotonated hydroxyl groups adsorb on the AgNP surface [42]. The AgNP surface increases its negative charge by this adsorption [42]. Electrostatic interactions between oxygen atoms in carbonyl- and hydroxyl-π-electrons
AgNP + R–C=O ⇆ R–C=O⋯⋯AgNP (3)
with the positively charged naked AgNP lead to adsorption onto the AgNP surface as well [42]. The same mechanisms are valid for the bioconstitutents absorbing around 320 nm in the LS-AgNP samples up to 6 days. Quercetin, chlorogenic acid, catechin, kaempferol, cirsimaritin, apigenin, luteolin, hesperidine, and thymol may act as capping and stabilizing agents by adsorbing on the AgNP surface through their negatively charged oxygen atoms [26,42,50].

#### 3.2.2. Fourier Transform Infrared (FTIR) Spectroscopy

FTIR analysis was employed to analyze the composition of the LS extract and the biocomposite LS-AgNP (Table 2, Appendix A).

The two FTIR spectra show similarities in most of the regions (Table 2, Appendix A). The phenolic acids from LS are rosmarinic, caffeic, cinnamic, *p*-coumaric, carnosic, and ferulic acid [35,42,43,44,72]. These compounds are represented in the FTIR absorption spectrum of both title compounds by the bands around 3500–3100, 2980, 2850, 1659, 1456, 1449, 1420, 1379, 1331, 1275, 1097, 1085, and 1050 cm^−1^ Table 2, Appendix A) [43]. Both spectra show broad bands around 3500 to 3100 cm^−1^ for the hydrogen bonded alcohol, carboxylic acid-OH, and amide NH groups originating from the biomolecules within LS Table 2, Appendix A). Carboxylate groups with their carbonyl C=O asymmetric stretching vibration appear in the spectrum around 1700 to 1600 cm^−1^ (Table 2, Appendix A). The pure LS extract shows very low absorption peaks at around 1663 cm^−1^ (Appendix A). LS-AgNP absorbs with slightly higher intensity at 1679 cm^−1^ (Table 2, Appendix A). The increase in absorption intensity combined with a shift to a higher wavenumber of 1679 cm^−1^ verifies the emergence of asymmetric carboxylate moieties. These groups appear due to the reduction of silver ions by deprotonated hydroxyl groups (Equation (1)). The symmetric stretching vibrations of the carboxylate groups absorb at 1419, 1449, 1456, and 1379 cm^−1^ (Table 2, Appendix A). These bands except the band at 1379 cm^−1^ absorb at lower intensities in the LS-AgNP in comparison to the pure LS (Table 2, Appendix A). When silver ions are reduced to AgNP,
Ag + R–COO^−^ ⇆ R–COO^−^;⋯⋯Ag (4)
these groups adsorb on the AgNP surface. The band at 1379 cm^−1^ became narrower (Appendix A). The Ag–OOC interaction is clearly verified and is in agreement with previous reports for the bands at 1419, 1449, 1456, and 1379 cm^−1^ [42,73].

The C–OH groups reveal absorption bands for the C–O stretching vibration at 1275, 1030 (phenols), around 1085 cm^−1^ (secondary alcohols), and 1050 cm^−1^ (primary alcohols) Table 2, Appendix A). The latter are showing higher absorption intensities in the LS-AgNP biohybrid (Table 2). The absorption band for the C–O stretching vibration at 1103 cm^−1^ in pure LS is found at 1094 cm^−1^ in LS-AgNP (Table 2, Appendix A). This band is accompanied by O–H stretching vibrations around 3350 cm^−1^ as well as two O–H bending at 1330 and 656 cm^−1^ (Table 2, Appendix A). The red shift from 1103 to smaller frequency and lower energy at 1094 cm^−1^ combined with the reduced absorption intensity suggests an increase in the mass of the concerned secondary alcohols by higher encapsulation and hydrogen bonding interactions. This is in agreement with previous reports of another Ag–O–C interaction on the silver surface by secondary deprotonated hydroxyl groups (Equation (2)). The absorption bands of the phenolic C–O stretching vibrations at 1275 cm^−1^ are both the same in intensity and appearance in both spectra Table 2, Appendix A). We can assume that the phenolic hydroxyl groups do not encapsulate the AgNP due to their bulky, hydrophobic phenolic groups. Instead, the carboxylate C–O and C=O are adsorbed, while the phenolic groups are pointing outwards [42]. The intensity in both figures remains the same, because the concentration of phenolic hydroxyl groups remains equal. Phenolic hydroxyl groups assist in reducing the silver ions, as seen in the reduced absorption intensity in LS-AgNP at 1030 cm^−1^ (Appendix A). This is confirmed by the decrease of conjugation around the aromatic ring CH-CH bonds through increased absorption intensity at 880 cm^−1^
Table 2, Appendix A). The same results were reported for the AV and Sage biomolecules in our previous investigation on AV-Sage formulations [43]. C=O groups are reduced back to C–OH in order to reinstate the stable conjugated system within the molecules and to maintain equilibrium.

The band at 1085 cm^−1^ is related to C–O stretching vibrations of ester and secondary alcohol groups (Table 2, Appendix A). The higher absorption intensity in LS-AgNP reveals that these groups are not encapsulating the AgNP and do not form hydrogen bonding (Table 2, Appendix A). The very small intensity band related to esters at 1730 cm^−1^ indicates a very low amount of intact rosmarinic acid in the sample. The very low absorption intensities for carbonyl C=O in esters confirm this assumption. The ester bond may have been broken to produce caffeic acid and 3,4-dihydroxyphenyllactic acid. The latter compound is the only polyphenol with a primary alcohol group.

The C–O stretching bands for primary alcohols appear around 1075–1000 cm^−1^ (Table 2, Appendix A). These confirm the availability of 3,4-dihydroxyphenyllactic acid and the solvent ethanol in the samples. After adding silver ions, the C–O stretching vibration absorption intensities for primary and secondary alcohols increase around 1045 to 1100 cm^−1^ (Appendix A). The increase may refer to reduced conjugation by reduction from C=O to C–O, less encapsulation, and reduced hydrogen bonding interactions. Primary alcohol groups are within ethanol and 3,4-dihydroxyphenyllactic acid. Ethanol molecules had dipol–dipol interactions and hydrogen bonding with the bioconstituents in the pure LS extract. After the AgNP synthesis, the biomolecules acted as reducing and capping agents. Ethanol molecules were removed from the biomolecules due to steric hindrance and crowding on the AgNP surface. Ethanol molecules were set free, while 3,4-dihydroxyphenyllactic acid is produced by breaking the ester bond in rosmarinic acid molecules.

The broad shoulder around 2750 cm^−1^ is related to an aldehyde H–C=O stretching vibration (Table 2, Appendix A). This band belongs to cinnamaldehyde in both of the samples according to previous reports (Figure 4) [42,74]. Cinnamaldehyde reduces silver ions and is oxidized to cinnamic acid [42].

As a conclusion, primary, secondary alcohol, and phenolic hydroxyl groups do not adsorb readily on the AgNP surface. The reduction of silver ions happens through cinnamaldehyde and phenol OH-groups producing cinnamic acid and carbonyl groups, respectively (Equation (1)). The increase of C=O groups is confirmed by the absorption band with increased intensity at 1659 cm^−1^ (Figure 4b). The adsorption to the metallic surface happens through the carboxylate groups and the carbonyl C=O entities in the biomolecules (Equations (2)–(4)) [42]. This hints at the availability of hydroxycinnamic acids (caffeic, cinnamic, *p*-coumaric, ferulic, chlorogenic, sinapic acid). These molecules form a layer on the AgNP surface with the carboxylate groups adsorbed on the metallic silver surface and the phenol groups pointing outwards. The encapsulation stabilizes the AgNP and prevents agglomeration. Silver ions are released during this process, which reinstates the concentration and stability of the resonance system of the phenolic acid groups during equilibrium.

#### 3.2.3. X-ray Diffraction (XRD)

The examination of composition and purity of LS-AgNP was investigated by XRD analysis. The spectrum reveals peaks at 2θ = 23.3°, 27°, 27.6°, 29.2°, 29.6°, 31.7°, 32°, and 46° (Figure 3).

**Figure 3 pharmaceutics-13-01352-f003:**
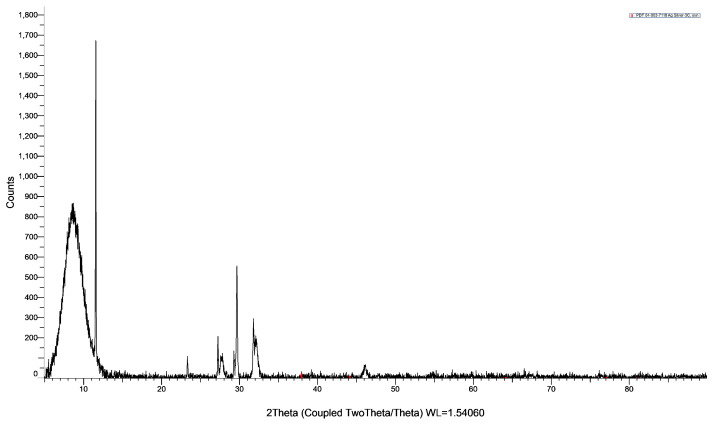
X-ray diffraction (XRD) analysis of LS-AgNP.

The XRD spectrum of LS-AgNP shows sharp peaks around 2θ = 23.3° (001), 27° (001), 29.6° (003), 31.7° (003), and 46° (200) (Figure 3). The broad bands from 2θ = 0° to 13°, 27° to 28°, and 32° to 33°, as well as the peak at 12° are due to the biomolecules in LS (Figure 3). These broad bands reveal amorphous phases related to the biocomponents of LS, as mentioned in previous reports [42]. The peaks around 27.6° and 29.2° originate from AgCl (Figure 3) [19,20,21,35]. Oxidation of AgNP, moisture, the equilibrium process itself, and degradation of the capping/stabilizing agents adsorbed on the AgNP surface cause the release of silver ions [18,29,42]. Cl ions are available in the plant material itself and could be the result of uptake from our UAE soil during plantation. The sharp peaks confirm the presence of crystalline phases in agreement with investigations on other bio-synthesized AgNPs [13,21,28,35,42]. The biohybrid LS-AgNP incorporates semicrystalline morphology with amorphous phases originating from LS biocomponents.

The XRD analysis of LS-AgNP verifies by the lack of other phases the purity of the nanocolloid (Figure 3).

#### 3.2.4. Antimicrobial Activities of LS-AgNP

The phenomenon of resistance is aggravating the need to develop new antimicrobials for a better future. The survival of mankind depends on new strategies to combat nosocomial infections, morbidity, and mortality due to unsuccessful treatment [1,2,3,4]. These factors impacted severely ill patients and caused unnecessary suffering, treatment failure, and fatal outcomes during the COVID-19 pandemic [2,3,4].

Investigations on the antimicrobial activities of NP and phytochemicals are raising steadily the benchmarks to achieve the set targets [8,9,10,11,12,13,14,18,19,20,21]. Silver ions released from AgNP are known to inhibit especially Gram-negative microorganisms due to their interaction with different parts of the bacterial cells [18]. They prevent protein synthesis, DNA replication, and biofilm formation by entering through the porin channels of Gram-negative pathogens [18]. Plant phytochemicals such as polyphenols, hydroxy-cinnamic acids, flavonoids, and further constituents have inhibitory action against microorganisms [8,9,10,11,23,24]. Phytochemicals are a rich source of natural antimicrobials, which can be utilized as microbicides [24]. Such molecules can enhance the antimicrobial properties of any formulation [24]. In this work, we reported the effect of LS extract on the antimicrobial action of AgNP.

The antimicrobial properties of LS-AgNP were investigated by agar well (AW) and disc diffusion assay (DD) against 10 reference microorganisms. The concerned Gram-positive strains included *S. pneumoniae* ATCC 49619, *S. aureus* ATCC 25923, *S. pyogenes* ATCC 19615, *E. faecalis* ATCC 29212, and *B. subtilis* WDCM 0003. The tested Gram-negative bacteria were *E. coli* WDCM 00013 Vitroids, *P. mirabilis* ATCC 29906, *P. aeruginosa* WDCM 00026 Vitroids, and *K. pneumoniae* WDCM 00097 Vitroids. The yeast *C. albicans* WDCM 00054 Vitroids was also utilized against LS and LS-AgNP. Gentamicin and nystatin were used as positive control antibiotics for bacterial strains and the fungus, respectively. The results of the antibiotics were compared to the inhibitory action of LS and LS-AgNP by measuring the zone of inhibition (ZOI) in mm. Ethanol and water were utilized as negative controls and showed no zone of inhibition (ZOI). The results of the negative controls are not mentioned in any table. The results of agar well (AW) and disk dilution (DD) studies of pure LS, AW of 10% LS-AgNP at pH 8.5 (10-AW), and AW of 10% LS-AgNP at pH 7 (10-AW-7) are represented in Table 3.

The pure LS extract inhibits in AW studies only three Gram-positive pathogens intermediately, while the rest of the tested pathogens are resistant (Table 3). DD studies at concentrations of 0.01 g/mL reveal higher antibacterial action against the Gram-negative *P. aeruginosa* (17 mm), which is followed by *E. coli* (15 mm) and *K. pneumoniae* (14 mm) (Table 3, Figure 4a–c).

**Figure 4 pharmaceutics-13-01352-f004:**
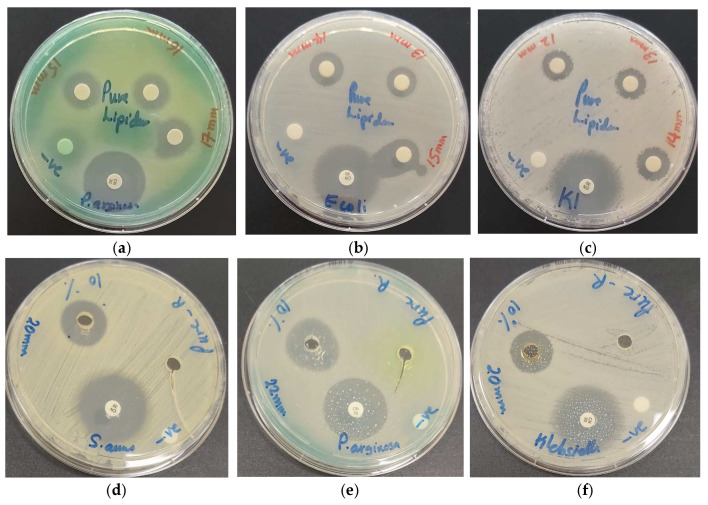
Susceptibility of reference strains against LS pure (0.01, 0.005, 0.0025 g/mL) and LS-AgNP (10%, 1.08 µg/mL) by disc diffusion assay (DD) and agar well method (AW) with positive control antibiotic gentamicin (30 µg/disc). From left to right: DD LS against (**a**) *P. aeruginosa* WDCM 00026; (**b**) *E. coli* WDCM00013; (**c**) *K. pneumoniae* WDCM 00097; AW LS and LS-AgNP against (**d**) *S. aureus* ATCC 25932; (**e**) *P. aeruginosa* WDCM 00026; (**f**) *K. pneumoniae* WDCM 00097.

The bioconstituents caffeic, ferulic, and cinnamic acid as well as other hydroxycinnamic acids are not supported by the AW method due to their low solubility in water. The AW method diminishes the antimicrobial action of the hydroxycinnamic acids against most of the utilized microorganisms due to their hydrophobic properties (Table 3, Figure 4d–f).

The DD method is more suitable for the pure LS extract (Table 3). Gram-negative pathogens were more inhibited than Gram-positive (Table 3). Gram-positive pathogens showed intermediate results with *S. aureus* (14 mm), which is followed by *B. subtilis* (13 mm), *S. pneumoniae* (12 mm), and *S. pyogenes* (11 mm) at a concentration of 0.01 g/mL (Table 3). Pure Sage extract with its bioconstituents did not inhibit Gram-positive bacteria in previous investigations at a concentration of 1 g/mL [42]. The tests revealed antibacterial action against the Gram-positive microorganisms *S. aureus* (13 mm), *B. subtilis* (13 mm), and *S. pyogenes* (11 mm) [43]. The pure LS extract was more efficient against the selected bacteria in comparison to Sage extract [43].

The AW method augmented the antimicrobial properties of the 10% LS-AgNP (pH = 8.5) compared to all other tests (Table 3). The highest inhibition was seen in *P. aeruginosa* (22 mm), which is followed by *K. pneumoniae* (20 mm), *S. aureus* (20 mm), *B. subtilis* (18 mm), *E. coli* (17 mm), *S. pneumoniae* (15 mm), *S. pyogenes* (15 mm), *E. faecalis* (15 mm), and *C. albicans* (13 mm) (Table 3, Figure 4d–f). Pure LS had no inhibitory action against Gram-negative pathogens in AW studies. The inhibitory action in AW seems to depend on the release of silver ions from LS-AgNP [18]. The best results in this study were recorded for the AW method of 10% LS-AgNP (pH = 8.5), which was followed by the DD method of the same formulation. Further DD studies revealed decreasing inhibitory action in the following order: pure LS, 5% LS-AgNP (pH = 8.5), 1-week-old 10% LS-AgNP (pH = 8.5), and lastly 10% LS-AgNP (pH = 7) (Table 4).

The DD studies of 10% LS-AgNP (pH = 8.5) show the highest inhibition zones for the Gram-negative *P. aeruginosa* (20/18/15 mm), which is followed by *E. coli* (15/14/13 mm) and *K. pneumoniae* (15/14/12 mm) (Table 4, Figure 4). The Gram-positive pathogens can be arranged with slightly lower results in the order *S. pneumoniae* (15/13/12 mm), *S. aureus* (14/13/12 mm), *S. pyogenes* (13/12/11 mm), and *E. faecalis* (13/12/11 mm) (Table 4, Figure 5).

Other groups reported recently similar results with plant biosynthesized AgNP [13,25,26,27,28,29]. Gram-positive and Gram-negative pathogens are susceptible to reported AgNP biohybrids [13,25,26,27,28,29]. DD studies showed similar zones of inhibition for *S. aureus* (13/14 mm), *E. coli* (12 mm), and *P. aeruginosa* (9 mm) [13]. AgNP synthesis by *Aaronsohnia factorovskyi* revealed antibacterial action against *S. aureus* (19 mm), which is followed by *P. aeruginosa* (15.33 mm), *E. coli*, and *B. subtilis* with 13.83 mm in descending order [28]. Cinn-AgNP showed much lower antimicrobial properties compared to LS-AgNP at the same concentration with the same reference strains [42]. The susceptible bacteria were *E. coli* (11 mm), *P. aeruginosa* (10 mm), and *S. aureus* (10 mm) [42]. The LS extract ameliorated the antimicrobial properties of AgNP in comparison to Cinn bark extract [42].

DD studies with LS-AgNP synthesized at pH = 8.5 of fresh 5% and 1-week-old 10% samples showed lower inhibitory action on the selected Gram-positive pathogens except *S. aureus* (Table 4). The 10% LS-AgNP synthesized at pH 7 inhibited the microorganisms intermediately at concentrations of 1.08 µg/mL and 0.54 µg/mL (Table 4). The pathogens were resistant at a concentration of 0.27 µg/mL (Table 4). These results are in agreement with recent investigations on the time, pH, and concentration dependence of AgNP [26]. Our samples did not inhibit *C. albicans* except in the AW study of 10% LS-AgNP (pH = 8.5) and the DD study of 10% LS-AgNP (pH = 8.5).

As a conclusion, the antimicrobial action of LS-AgNP is concentration-, time- and pH-dependent. Our samples inhibited all the selected reference strains under given conditions. Remarkable results are the broad spectrum antimicrobial activity against ESKAPE pathogens. Best results were achieved by 10% LS-AgNP at a concentration of 1.08 µg/mL, synthesized with a 10% content of silver ions, at pH 8.5 and not stored longer than 6 days.

The increased number of adsorbed biomolecules did not enhance the antimicrobial properties in any of the cases except for the 10% LS-AgNP formulation for up to 6 days. The best antimicrobial results were achieved by fresh 10% LS-AgNP prepared at a pH of 8.5. The same finding was presented by Fanoro et al. recently [17]. AgNP releases silver ions during the equilibrium process by oxidation and increases the antimicrobial properties of the LS-AgNP [17,18,21]. AgCl molecules are confirmed by the XRD analysis (Figure 5). The pure LS extract showed no antimicrobial action in AW studies against the Gram-negative pathogens. This finding confirms the antimicrobial action of released silver ions in the AW studies of LS-AgNP. The released silver ions move through the porin channels of the Gram-negative bacteria and exert their antibacterial action. The plant biocompounds are not able to move through the porin channels in AW studies.

Gram-negative pathogens were more susceptible. Noteworthy is the high inhibition of the multidrug-resistant *P. aeruginosa*. The zone of inhibition was 22 mm in AW studies of 10% LS-AgNP compared to the antibiotic gentamicin with 23 mm (Table 3). The multiple drug efflux systems of these rod shaped, motile bacilli were highly susceptible against all our samples except in the AW study of pure LS.

*P. mirabilis* is the only resistant Gram-negative bacteria in this series. It is a rod-shaped bacillus with multiple *flagellae* and swarming motility. This morphology enables *P. mirabilis* resistance against our title compounds except in both AW studies of 10% LS-AgNP at pH 8.5 and 7 with ZOI = 10 mm (Table 3).

The title compounds exhibit the highest antimicrobial properties against rod-shaped, motile bacilli, which is followed by round *strepto-* and *staphylococci*. The round-shaped cocci can be ordered according to the impact of our samples and their morphology. Clustered cocci are more susceptible than chains and pairs of bacteria.

Gram-negative bacteria with their small peptidoglycan layers, less crosslinking, and highly negatively charged outer cell membranes with lipopolysaccharides are more susceptible due to their porin channels [8,44,72]. Silver ions, AgNP, smaller, lipophilic hydroxycinnamic acids and further phytochemicals move through these channels or puncture the outer cell membrane [8,13,18,42,43,44]. They disrupt protein synthesis, inhibit DNA replication, cancel efflux systems, and result in cell death [8,13,18,22,23,24,28,42,43,44,75].

Gram-positive bacteria are inhibited by our title compounds as well. These pathogens consist of a lower negatively charged outer cell membrane. The Gram-positive bacteria have a thick peptidoglycan layer crosslinked by peptides and inclusions of lipoteichoic as well as teichoic acid [8,44,72]. The phytochemicals in our samples interact with the partial negatively charged atoms in the cell membranes and crosslinked peptides to destabilize the cell wall structure by intermolecular interactions [43]. Phytochemicals prevent bacterial attachment on mucosal surfaces, reduce biofilm formation, inhibit quorum sensing, influence cell surface hydrophobicity, and reduce glycolytic enzymes [23,24].

## 4. Conclusions

Increasing cases of microbial resistance to drugs and antimicrobials endanger the future of mankind. Plants developed mechanisms to defend themselves against harmful pathogens since the beginning of their existence. Many civilizations utilized phytochemicals in medicinal plants, herbs, spices, and agricultural products as antimicrobial agents. Silver is known and has been used for centuries by mankind against microorganisms. There is an increasing number of publications on phytochemicals and the plant-based synthesis of silver nanoparticles (AgNP). We investigated the antimicrobial properties of the widespread plant LS, which is also known as curly garden cress. Our aim was to ameliorate these properties by utilizing the LS in the green synthesis of AgNP. The combination of LS with AgNP enhanced the inhibitory action against notorious ESKAPE pathogens. The LS-AgNP bio-composites revealed pH, time, and concentration dependencies related to their antimicrobial action. The stability and antimicrobial properties are directly related to the availability of conjugated systems within the bio-compounds and inversely related to agglomeration. LS phytochemicals acted as reducing, stabilizing, and capping agents for AgNP. Carbonyl C=O groups adsorbed on the metallic silver surface and encapsulated the AgNP, resulting in small sizes around 36.1 nm. Silver ions are released by AgNP into the colloidal suspension due to the equilibrium, moisture, and oxidation of AgNP, as well as degradation of the plant-based capping agents. Silver ions increase the inhibitory action against Gram-negative bacteria by moving through their porin channels. Pure LS extract and the 10% LS-AgNP formulations at concentrations of 1.08 µg/mL, 0.54 µg/mL, and 0.27 µg/mL showed very good to intermediate antimicrobial actions on the selection of 10 microorganisms. Gram-negative strains of *P. aeruginosa*, *K. pneumoniae*, and *E. coli* are susceptible, as well as Gram-positive *S. aureus*, the spore-forming *B. subtilis*, *S. pneumoniae*, *S. pyogenes*, and *E. faecalis*. Antifungal activity against *C. albicans* was detected in agar well and disc diffusion studies of two of our 10% LS-AgNP samples. The results confirm the potential uses of LS-AgNP as broad-spectrum microbicides, disinfectants, and wound care products.

## Figures and Tables

**Figure 1 pharmaceutics-13-01352-f001:**
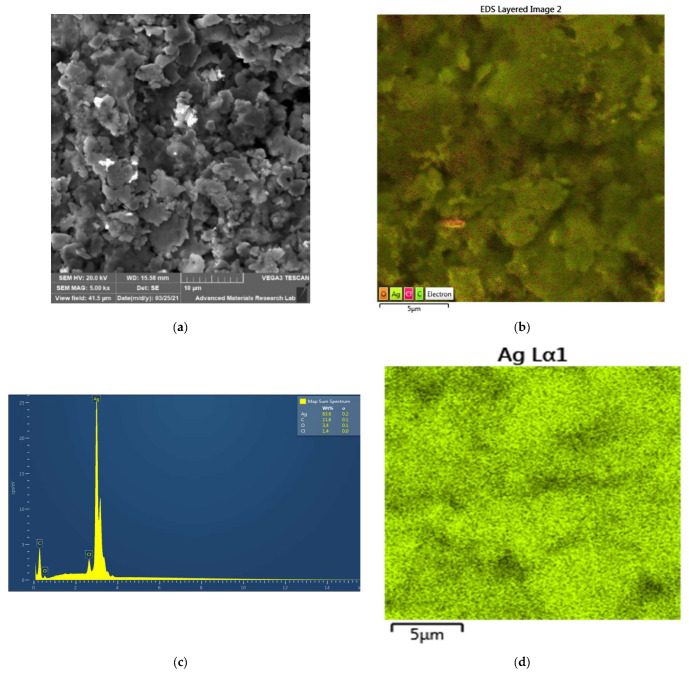
Scanning electron microscopy (SEM) (**a**); energy-dispersive spectroscopy (EDS) (**b**); layered EDS (**c**); EDS analysis; and (**d**) layered EDS of LS-AgNP.

**Figure 2 pharmaceutics-13-01352-f002:**
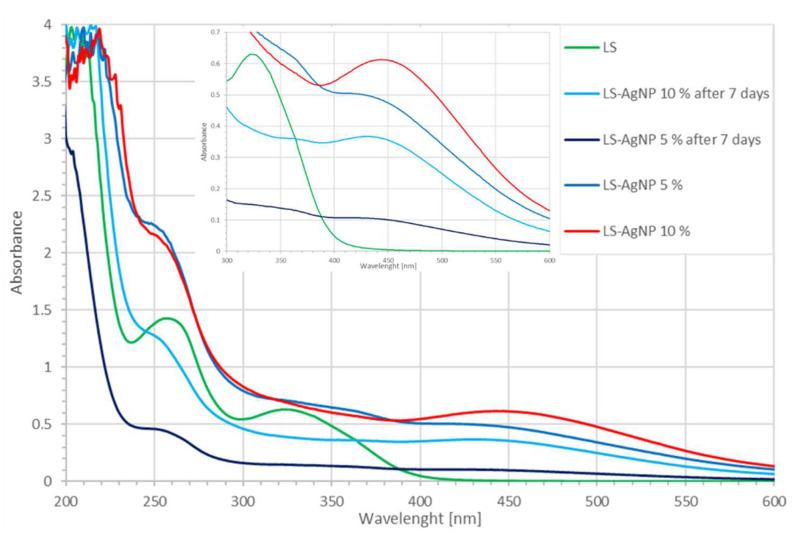
UV-vis analysis of LS and LS-AgNP: with inset of 300–600 nm (LS extract: dark green; LS-AgNP: dark blue; 5% LS-AgNP: black; 5% LS-AgNP after 7 days: red; 10% LS-AgNP: light blue; 10% LS-AgNP after 7 days).

**Figure 5 pharmaceutics-13-01352-f005:**
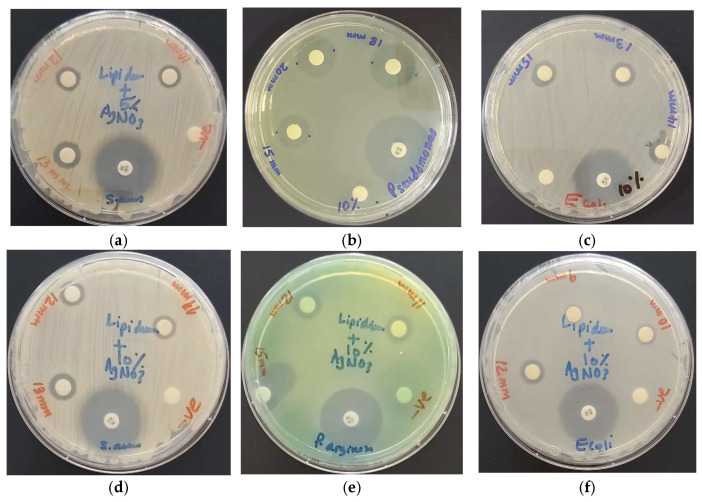
LS-AgNP_._ with positive control antibiotic gentamicin. From left to right: (**a**) 5% LS-AgNP (0.54 µg/mL, 0.27 µg/mL, and 0.14 µg/mL) against *S. aureus* ATCC 25932; 10% LS-AgNP (1.08 µg/mL, 0.54 µg/mL, and 0.27 µg/mL) against: (**b**) *P. aeruginosa* WDCM 00026; (**c**) *E. coli* WDCM 00013; (**c**) *E. coli* WDCM 00013; 1-week-old, aged 10% LS-AgNP (1.08 µg/mL, 0.54 µg/mL and 0.27 µg/mL) against (**d**) *S. aureus* ATCC 25932; (**e**) *P. aeruginosa* WDCM 00026; (**f**) *E. coli* WDCM 00013.

**Table 1 pharmaceutics-13-01352-t001:** Zeta potential measurements and DLS results for each sample of LS-AgNP.

Sample	Zeta Potential (mV)	Particle Size Mean (nm)	Z-Average (nm)	Polydispersity Index (PDI)
5% LS-AgNP	−0.1	20.3 ± 11.5	65.5	0.735
10% LS-AgNP	−0.2	36.1 ± 12.3	57.9	0.388

**Table 2 pharmaceutics-13-01352-t002:** Fourier transform infrared (FTIR) spectroscopic analysis of LS and LS-AgNP (cm^−1^).

Group	LS	LS-AgNP
(O–H)ν	3100–3600	3100–3600
(H-C=O)ν	2750	2750↑
(O–H)ν	2990	2980↑
	2850	2850
(C=O)ν_as_	1663	1679
(C=O)ν_as_		1659
(C=O)ν_s_	1418	1419↓
	1448	1449↓
	1458	1456↓
	1385, 1379	1379
(C–O–C)ν_s_	1330	1330
(C–OH)ν (C–O)ν	1103	1097↓
	1275	1275
	1085	1085↑
	1050	1050↑
	1030	1030↓
	804	804↑
	656	656↑
(CH–CH)ν	880	880↑

ν = vibrational stretching, s = symmetric, a = asymmetric. ↑ = increased absorption intensity, ↓ = reduced absorption intensity.

**Table 3 pharmaceutics-13-01352-t003:** Antimicrobial testing of antibiotics (A), LS by disc dilution studies (1, 2, 3), LS (LS-AW), 10% LS-AgNP at pH 8.5 (10-AW), and 10% LS-AgNP at pH 7 (10-AW-7) by agar well method. Microbial strain susceptibility indicated by ZOI (mm).

Strain	Antibiotic	A	LS1 ^+^	LS2 ^+^	LS3 ^+^	LS-AW	10-AW	10-AW-7
*S. pneumoniae* ATCC 49619	G	18	12	11	10	12	15	14
*S. aureus* ATCC 25923	G	28	14	13	13	0	20	19
*S. pyogenes* ATCC 19615	G	25	11	10	9	10	15	14
*E. faecalis* ATCC 29212	G	25	0	0	0	12	15	14
*B. subtilis* WDCM 00003	G	21	13	12	10	0	18	18
*P. mirabilis* ATCC 29906	G	30	0	0	0	0	10	10
*P. aeruginosa* WDCM 00026	G	23	17	16	15	0	22	22
*E. coli* WDCM 00013	G	23	15	14	13	0	17	17
*K. pneumoniae* WDCM 00097	G	30	14	13	12	0	20	20
*C. albicans* WDCM 00054	NY	16	0	0	0	0	13	12

^+^ Disc diffusion studies (6 mm disc impregnated with 2 mL of 0.01 g/mL (LS1), 2 mL of 0.005 g/mL (LS2), and 2 mL of 0.0025 g/mL (LS3) of LS. LS-AW = Agar well method with LS extract of 0.01 g/mL. 10-AW = Agar well method with 10% LS-AgNP (1.08 µg/mL) at pH = 8.5. 10-AW-7 = Agar well method with 10% LS-AgNP (1.08 µg/mL) at pH = 7. A = G Gentamicin (30 µg/disc). NY (Nystatin) (100 IU). The gray shaded area represents Gram-negative bacteria. 0 = Resistant. No statistically significant differences (*p* > 0.05) between row-based values through Pearson correlation.

**Table 4 pharmaceutics-13-01352-t004:** Antimicrobial testing by disc diffusion (1, 2, 3) of antibiotics (A), LS-AgNP (5 and 10%, pH = 7 and 8.5, up to 5 days to 1-week-old sample). Microbial strain susceptibility indicated by ZOI (mm).

Strain	Antibiotic	A	5-1 ^+^	5-2 ^+^	5-3 ^+^	10-1 ^+^	10-2 ^+^	10-3 ^+^	10 *-1 ^+^	10 *-2 ^+^	10 *-3 ^+^	10-7-1 ^+^	10-7-2 ^+^	10-7-3 ^+^
*S. pneumoniae* ATCC 49619	G	18	10	0	0	15	13	12	11	0	0	14	13	0
*S. aureus* ATCC 25923	G	28	13	12	10	14	13	12	13	12	11	12	10	0
*S. pyogenes* ATCC 19615	G	25	11	10	9	13	12	11	11	9	0	11	10	0
*E. faecalis* ATCC 29212	G	25	0	0	0	13	12	11	9	0	0	0	0	0
*B. subtilis* WDCM 00003	G	21	9	0	0	13	11	10	10	0	0	14	13	0
*P. mirabilis* ATCC 29906	G	30	0	0	0	0	0	0	0	0	0	0	0	0
*P. aeruginosa* WDCM 00026	G	23	15	14	12	20	18	15	15	12	11	18	15	0
*E. coli* WDCM 00013	G	23	14	13	9	15	14	13	12	10	9	16	15	0
*K. pneumoniae* WDCM 00097	G	30	14	13	10	15	14	12	13	12	11	15	11	0
*C. albicans* WDCM 00054	NY	16	0	0	0	0	0	0	0	0	0	14	0	0

^+^ Disc diffusion studies (6 mm disc impregnated with: 5% LS-AgNP 2 mL of 0.54 µg/mL (5-1), 2 mL of 0.27 µg/mL (5-2) and 2 mL of 0.14 µg/mL (5-3). 10% LS-AgNP 2 mL of 1.08 µg/mL (10-1), 2 mL of 0.54 µg/mL (10-2), and 2 mL of 0.27 µg/mL (10-3). 10% LS-AgNP 2 mL of 1.08 µg/mL at pH = 7 (10-7-1), 2 mL of 0.54 µg/mL (10-7-2), and 2 mL of 0.27 µg/mL (10-7-3). * One-week-old 10% LS-AgNP 2 mL of 1.08 µg/mL (10 *-1), 2 mL of 0.54 µg/mL (10 *-2), and 2 mL of 0.27 µg/mL (10 *-3). A = G Gentamicin (30 µg/disc). NY (Nystatin) (100 IU). The gray shaded area represents Gram-negative bacteria. 0 = Resistant. No statistically significant differences (*p* > 0.05) between row-based values through Pearson correlation.

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
