# Peer review of "Antimicrobial Properties of Lepidium sativum L. Facilitated Silver Nanoparticles"

_pharmaceutics, 2021, doi:10.3390/pharmaceutics13091352_

Round 1

Reviewer 1 Report

The article entitled "Antimicrobial properties of Lepidium satvium L. facilitated Silver Nanoparticles" presents a fascinating look at the process of green synthesis of bionanomaterials. Unfortunately, in its current form, despite a very nice analysis of the results, the article requires many corrections:

- line 109: DLS does not confirm the composition

- lines 106-117: information on XRD and SEM studies and the purpose of these methods is missing

- lines 236-243: I do not understand the purpose of these sentences at this point of the article

- line 261: “unimodal size distribution” of what? it is not explained precisely

- lines 276-283: there is no information about the origin of Cl, or structural formulas of individual compounds. What arises and why do they are interacting with Ag?

- line 291: it should be spectroscopy

- lines 302 and 305-306: not precisely defined "smaller". it is worth clarifying the size

-line 309-312 or 446-447: there should be no information about antibacterial properties

- chapters 2.2 and 2.3 should be illustrated with a picture showing the synthesis process step by step

- Fig. 1, the proportions between individual panels are strongly disturbed, making it difficult to perceive the drawing. I would suggest including panels a and c in the foreground, and treat panels b and d as addition and present them in a reduced size. Image d also shows the distribution of Ag in a specific fragment of space. However, it suggests that the silver is distributed throughout the volume. However, nothing is known about its size, shape, or degree of aggregation. At this point, it would be helpful to show TEM photos showing the behavior of Ag after the synthesis process for both 5% and 10% samples.

- fig. 2 and 4 look like screenshots. I would suggest preparing them in external programs taking into account xy data. How was the size of the Ag nanoparticles obtained? No statistical information. What does Figure 2b show? There is no comparison between the 5% and 10% samples. Where do the differences in the size of nanoparticles in lines 262 and 269 come from?

- fig. 4, IR spectra seem to be measured inappropriately. It seems that there were measured in absorbance mode with the too thick sample. Therefore the bands are cut, or there are very rough. According to the software proposition, there are also no bands in regions 1600-1800 or 600-800cm-1. In Supplementary files, tables are not necessary because there were not discussed in the text.  

- fig. 5, during the description of the XRD results, the formation of AgCl was shown, while it is nowhere commented on or discussed. Please explain that this phase comes from, what is important in the context of antimicrobial properties, and how the described mechanism of conjugate bond formation or the formation of a layer surrounding Ag translates into this phase. How the XRD looks of pure LS and the comparison of 5% and 10% samples

Reviewer 2 Report

Article Edis et al. "Antimicrobial properties of Lepidium satvium L. facilitated Silver Nanoparticles" is devoted to the synthesis, physicochemical characterization, and assessment of antibacterial and antifungal properties of bio-nanohybrid AgNPs. The authors fabricated AgNPs by reducing silver nitrate in L. satvium extracts. The components of the extract served as reducing and stabilizing agents. Were obtained stable hybrid bio-nanoparticles of approximately the same diameter of about 35 nm, containing crystal structures of silver associated with components of the extract. A comprehensive analysis of the obtained hybrid AgNPs was carried out carefully and accurately.

The antibacterial activity of hybrid AgNPs was carried out on a reference group of gram-positive and gram-negative bacteria, as well as on yeast, in comparison with antibiotics used against reference bacteria in the clinic, and LS extracts with antibacterial activity. It has been convincingly shown that LS-AgNPs exhibit a synergistic antibacterial effect. LS-AgNPs acted in dose, time and Ph dependence.Hybrid AgNPs were effective against yeast, which showed resistance to AgNPs and LS extracts.

The design of the work, the methods used, and the results obtained have no comments.

The work is aimed at finding antibacterial drugs with the properties of antibiotics, but not causing the development of resistance to them. Many teams are working in this direction. Do the authors admit that bacteria can develop resistance to AgNPs that form Ag(+) as well as to other metal ions?

Minor remark.

Section 3.1. called "Elemental Composition, Morphological Examination", the description of the results begins with morphology (SEM). I recommend swapping "a" and "b" in Figure 1.

Author Response

Thank you so much

Reviewer 3 Report

This paper reports the antimicrobial properties of Lepidium satvium L. facilitated Silver Nanoparticles. 

The materials are well characterized and show promising properties with potential  uses as broad spectrum microbicide, disinfectant and wound care product. In my opinion, the research is well done, and the conclusions are well supported by the data. 

On the other hand, the paper reads poorly, and it is not necessarily due to bad English (although some improvements are needed). 

Major revisions are necessary.

 I suggest the authors to put their work in a better context to the state-of-the-art about “green” syntheses of AgNPs from plant extracts or biomass.

Please provide a DLS analysis of the samples to study the behaviour of the suspensions in liquid.

What is the antimicrobial mechanism of the investigated samples?

Author Response

Thank you so much

Round 2

Reviewer 1 Report

I would like to thank the authors for the answer. Most of my comments have been accentuated while some of the points have not been completed or corrected as suggested. Unfortunately, the "vacation period" or "lack of appropriate software" is not the best explanation for me. The article should provide valuable data of the highest quality, both texts, and graphics. Therefore, one more please look at the doubts:

- chapters 2.2 and 2.3 should be illustrated with a picture showing the synthesis process step by step

- Fig. 1, the proportions between individual panels are strongly disturbed, making it difficult to perceive the drawing. I would suggest including panels a and c in the foreground, and treat panels b and d as addition and present them in a reduced size. Image d also shows the distribution of Ag in a specific fragment of space. However, it suggests that the silver is distributed throughout the volume. However, nothing is known about its size, shape, or degree of aggregation.

- fig. 2 and 4 look like screenshots. I would suggest preparing them in external programs or if not you can correct the quality of the figure using external graphical programs like Corel, photoshop, or gimp. How was the size of the Ag nanoparticles obtained? No statistical information. Where do the differences in the size of nanoparticles in lines 262 and 269 come from?

- fig. 4, IR spectra seem to be measured inappropriately. It seems that there were measured in absorbance mode with the too thick sample. Therefore the bands are cut, or there are very rough. According to the software proposition, there are also no bands in regions 1600-1800 or 600-800cm-1. In Supplementary files, tables are not necessary because there were not discussed in the text. 

Author Response

Dear Reviewer,

thank you for your comments.

Brest regards

Zehra

Reviewer 3 Report

Accept in present form.

Author Response

Thank you so much for your kindness.

Best regards

Zehra

Round 3

Reviewer 1 Report

Dear authors,

As for the pictures, honestly, I will remove figure 2 leaving only table 1 because is more readable, and remove picture 4 given in a table band position with their interpretation. I suggest also correct the quality of picture 1 according to my suggestion because you can do that even in paint.

Best Regards

Author Response

Dear Reviewer,

thank you very much for your sincere efforts to improve our manuscript. We are really grateful for your supportive advises.

Please see the attached pdf

With my best regards

Zehra
